# “He Just Doesn’t Want to Get Out of the Chair and Do It”: The Impact of Apathy in People with Dementia on Their Carers

**DOI:** 10.3390/ijerph18126317

**Published:** 2021-06-11

**Authors:** Chern Yi Marybeth Chang, Waqaar Baber, Tom Dening, Jennifer Yates

**Affiliations:** 1Clinical Neurosciences and Mental Health, School of Medicine, University of Nottingham, Nothingham NG7 2UH, UK; marybethchang98@gmail.com (C.Y.M.C.); tom.dening@nottingham.ac.uk (T.D.); 2The National Hospital for Neurology and Neurosurgery, University College London Hospitals NHS Foundation Trust, London WC1N 3BG, UK; waqaar.baber@nhs.net

**Keywords:** dementia, carers, apathy, motivation, activities, relationships, qualitative, burden

## Abstract

Apathy, defined as a lack of motivation, is a prevalent and persistent behavioural and psychological symptom of dementia. Limited research suggests that apathy is associated with increased carer burden, but there are no studies investigating carers’ subjective experiences of apathy. This study aimed to fill this gap and explore the lived experience of apathy in dementia from the perspectives of the people with dementia and their carers. This article reports on the carers’ perspectives. Six dyads of people with dementia and carers participated in semi-structured interviews, which were analysed using interpretative phenomenological analysis. Three superordinate themes were identified: (1) achieving a balance of conflicting emotions—the challenges of apathy led to feelings of guilt, acceptance, and frustration; (2) new roles imposed by caring, which involved taking on new responsibilities and promoting remaining interests of person with dementia; and (3) having a life of one’s own—coping with apathy by talking to others, and spending time away from the caring role. This study highlighted that carers are caught in a struggle between wanting to involve the person with dementia in decisions and finding that they cannot if they want to overcome the hurdle of apathy. Implications of this study suggest that a wider understanding of apathy at a societal level could lead to the provision of a helpful forum for carers to share their experiences.

## 1. Introduction

Apathy is characterised as a lack of motivation and decline of goal-directed behaviour compared to prior functioning [1]. Within dementia, it is considered a behavioural and psychological symptom (BPSD) and shows stability over time, suggesting that apathy is maintained for most people [2]. Apathy has high prevalence, with recent estimates suggesting it is experienced by approximately 49% of people with dementia [3].

Traditionally, apathy is considered the result of neurodegenerative disorder [4] and disease progression, such as cortical changes in the anterior cingulate cortex [5], a region linked to impaired executive functioning in dementia [6]. Apathy could result from environmental or external processes; associations between strength and clarity of environmental stimulation and apathy have been observed in care residents with dementia [7]. Stimulating environments may mitigate apathy. During high-intensity exercise sessions, high levels of motivation for people with dementia were found irrespective, of reported apathy [8]. Apathy is associated with disability, poorer health, and carer burden [2], due to a tendency to depend on others to initiate daily life activities, despite still being capable [9]. BPSDs generally have a negative impact on carers, including stress, inability to cope with accumulating tension, desperation, exhaustion over time, isolation, and relationship issues between the carer and person with dementia [10,11]. However, the effects of different BPSDs are often lumped together, and research that directly explores the impact of apathy in dementia on carers is lacking. Some published studies have explored this in other neurological conditions. For example, apathy in people with primary progressive aphasia (PPA) was a key contributor to carer burden [12], even after controlling for PPA variant, disease duration, and severity of cognitive and language impairments. Carers of people with amyotrophic lateral sclerosis (ALS) who reported greater behavioural symptoms of apathy exhibited by the people living with ALS had higher levels of burden, compared to those reporting lower levels of apathy [13]. Research suggests that rather than just the presence of BPSD in the person with dementia, it is the carer’s reaction that directly influences the carer’s BPSD-related distress. Psychosocial factors, including sense of competence, guilt, burden, and reactivity, may influence the course of BPSD, whereas the frequency of BPSD was not a significant predictor of carer distress [14]. Having a greater sense of confidence buffered against carer burden, whereas feelings of guilt were associated with lower quality relationships, potentially due to a decline in closeness between the person with dementia and their carer [15]. Burden may impact upon a carer’s ability to manage symptoms and lead to developing maladaptive strategies, such as reacting emotionally or avoiding challenging situations. Simpson et al. [16] explored the experiences of apathy in Parkinson’s disease (PD). Their findings suggested that apathy could be the psychological response to physical and functional impairments associated with PD, a threat to identity and an impairment. Differing private and public experiences of apathy present an alternative conceptualisation in contrast to traditional understandings of apathy as simply the consequence of an endogenous disease process, and instead conceptualise it as an internal struggle to maintain a sense of self that ultimately results in a behavioural manifestation of lack of motivation.

Previous research into apathy in neurological conditions has taken a quantitative and neurobiological approach. There are benefits to this, such as the development of pharmacological therapies [17], but there is a lack of consideration of how apathy affects the subjective wellbeing of carers. There is a discrepancy between the perspectives of carers and people with dementia, where carers rated apathy in people with dementia as worse over time, whilst self-ratings by people with dementia remained stable [18]. Carers’ responses and perceptions to apathy may contribute to their quality of life and wellbeing. Therefore, qualitative research is required to explore the lived experience of apathy in dementia, and particularly its impact on carers. No study to date has qualitatively explored the lived experience of the impact of apathy in dementia on carers.

This article reports an interpretative phenomenological analysis (IPA) of accounts of carers on their understanding of apathy in the person they care for, and the impact of apathy on both of them. Being mindful that most studies exploring the lived experience in dementia have involved proxy interviews with carers to access the experience of the person with dementia [19], this study has explored the phenomenon from both the perspective of the carer, detailed here, and also the person with dementia, which appears in a separate publication [20].

## 2. Methods

### 2.1. Design

A qualitative IPA approach was used. This approach emphasises how psychological meanings can be pursued through detailed examination of the lived experience [21] to explore how individuals makes sense of their experiences within a particular context. IPA “gives voice” to underrepresented and undervalued lived experience faced by carers of people with dementia [22,23]. Ethical approval was granted by the Faculty of Medicine & Health Sciences Research Ethics Committee, University of Nottingham (Ref: 449–1912). All participants provided written informed consent.

### 2.2. Sampling and Participant Recruitment

The overall study aimed to recruit 12 participants, 6 people with dementia and 6 carers. The analytic approach uses IPA; thus, “less is more” [24], where thoroughly exploring the experiences of fewer participants in greater depth is preferable [25]. Therefore, six interviews in each group honoured this principle. This paper reports the analysis of the six participants in the carers group.

Participants were recruited through engagement with local community dementia support groups, where a member of the research team spoke to group members about the study and invited interested people with dementia and carers to discuss further participation. Individuals who cared for a person with a diagnosis of dementia and were able to talk about their lived experience in English were eligible to participate. There were no restrictions on the type of dementia that the person with dementia had, the relationship between the carer and the person with dementia, or whether the carer or person with dementia explicitly self-identified as experiencing apathy.

### 2.3. Data Collection

In-depth, semi-structured interviews were used. An interview schedule guided the interview, with prompts regarding the subjective interpretation of apathy, impact of behaviour, habits, hobbies, relationships, mood, and activities in daily living. Interviews with carers and with people with dementia were conducted simultaneously in separate rooms by J.Y. and W.B., respectively. Separating the interviews enabled both people with dementia and carers to speak candidly about their experiences, and avoid the other person intervening [26,27]. Interviews lasted approximately one hour, were audio-recorded, and transcribed verbatim.

### 2.4. Data Analysis

Interviews were analysed using IPA guidelines [28], focusing on identifying the experiences of apathy in dementia and its impact on carers. Codes were identified by developing a close, line-by-line analysis of the experiential claims, concerns, and understandings of each of the six carer participants. Codes were listed alongside representative quotes for each transcript, and reflective or exploratory comments were added to develop meaning and understanding.

Each transcript was analysed individually, enabling the researcher to be open to new themes from each participant. After this process was completed for each transcript, a list of codes was collated to identify connections across the transcripts. Patterns in the codes across participants were explored, including commonalities and nuances within and between participants’ data to identify superordinate and subordinate themes. Themes were refined through discussing, confirming, and challenging the content of the themes by the research team.

## 3. Results

The six carers were all spouses, one male and five females, and were all White British, with a mean age of 69.3 years (range 59–84). The analysis identified three superordinate themes: (1) balancing conflicting emotions; (2) new roles imposed by caring; and (3) having a life of one’s own. The carers all recognised apathy in the person they cared for and described it as “*listless, about not having too much interest in things*”, or a “*lack of involvement or lack of enthusiasm to do something… lack of motivation*”. Carers drew upon particular examples to highlight how the person with dementia’s apathy crept into the everyday experience; for example, by no longer having a preference over what to eat, or not motivating themselves to eat without prompting, and neglect of personal hygiene: “*he just won’t go in the shower or a bath*”. There were various interpretations from the carers about what caused the apathy; for example, Carer 4 felt that her spouse was apathetic towards making choices because he simply could not “*be bothered to think*” and preferred someone else to make the effort to decide things on his behalf. Carer 3 recognised that her spouse “*procrastinates when there is a change*”, suggesting that new activities where the comfort of routine to guide the person with dementia could not be relied upon, and therefore the actions came to a stop. Carer 5 reflected that her spouse “*didn’t want confrontation*” and therefore preferred not to tackle particular tasks, especially administrative responsibilities. Participants in this study acknowledged that the apathy had changed the person they care for, and continued to explore the impact that it had on them.

### 3.1. Theme 1: Balancing Conflicting Emotions

The experience of apathy in the person with dementia created conflicting emotions for carers, because they had to negotiate between effectively tricking the person with dementia into a course of action, or allowing the apathy to persist, which was detrimental for both the person with dementia and the carer. Embarking on activities and outings enabled carers to maintain their support of the person with dementia through challenges and manage feelings of frustration regarding behavioural changes. Carers described strategies to manage apathy, which typically involved an element of deception and for which they felt guilty. Carers seemed caught between a rock and a hard place, where if they were open with the person with dementia about a potential course of action they would be refused, and no action would take place. If they deceived them, or were economical with the truth, they would both be able to enjoy and engage in activities, but this led to a feeling of guilt. Although carers felt strongly that the ends justified the means to create engaging opportunities for their spouses, the means created difficult emotional feelings. This was further complicated by a desire to include their spouses in decisions in the same way that they had throughout their lives together. Being able to accept the lack of involvement, or to ignore disinterest and refusals, required a sense of endurance. Negotiating this path was typically quite lonely for the carers, because their spouses often lacked insight regarding their apathy, or dementia more generally, and were unaware of the frustration that carers felt.

#### 3.1.1. Feelings of Guilt

Carer 2 explained that she found it easier to withhold information from her spouse to initiate activities or trips out that he would enjoy in order to avoid opportunities for him to decline, taking an approach where the ends justify the means even though it made her feel guilty. If he were unaware of the details, he would not question or feel concerned about the activity.


*“And these days I tend not to tell him too much when we go somewhere, I find he enjoys it… I know if I asked him he would say no, I am not bothered.”*


Carer 2’s spouse enjoyed himself during each trip; therefore, the tactic of withholding details was concluded to be “*a protection for him*” to ensure that he still engaged in activities. Being able to go out and do things benefitted them both:


*“I know that if we go it’s better for both of us, so I tend to not sort of supply too much information and just let it happen.”*


Even with the benefits this particular strategy brought, it made the carer feel uncomfortable, suggested by her several justifications of the strategy (“*it wouldn’t make any difference really whether he knew*”) during the interview. Perhaps at the heart of the issue is that the carer realised the similarity between this and caring for a child, acknowledging that she had to almost trick him into doing things, and showing that their relationship as partners had changed as her spouse took on a more childlike role:


*“…You are almost dealing with a child and I don’t like the, it’s not the deceit, deceit is probably too strong a word, but I am not being totally honest with him.”*


This carer, similar to other participants, found herself in a difficult position where she could either let him to sit in the house unengaged, or use a form of deception to allow him to participate in beneficial activities. Either course of action would result in feelings of guilt, but the carer chose the route that would lead to better outcomes for her spouse.

Carer 5 discussed similar strategies to motivate her spouse, acknowledging that she needed to be “*crafty*” to persuade him to do things by sowing the seed of an idea in the hope that he would come to believe it had been his own idea all along.


*“It’s like picking a colour of paint. We’ll pick a colour of paint and I’ll say I like this colour, and he says, no, I hate that colour. And then two weeks later he’ll like the colour, you know what I mean? But you have to play games, you know, that’s the sort of thing I have to do with him, I have to play games… something that I wanted but he was definitely we’re not having it, blah, blah, blah, no, no. We had a big domestic over it and then he agreed to it. He came up with the idea and I thought to myself, well, I thought of that two weeks ago, you know what I mean? And now you say you’re doing it.”*


A further balance between emotions for Carer 5 came from feeling cross about things not being done and not wanting to lose her temper and upset her spouse. She felt bad for nagging him, but also annoyed that important jobs, such as fixing her car, were not dealt with as they might have been prior to the apathy.


*“It takes that shove some—Sometimes I can push him too hard because then we have a domestic, you know. It’s just like eggshells sometimes. I have to do it slow-things. I can’t nag, nag, nag, nag, nag, you know, it just doesn’t—It’s just too much for him, I keep nagging. Sometimes I’ll lose my temper.”*


This carer also likened the experience to dealing with a child, reflecting that “*it’s like I’m telling off a baby, and I don’t want him to be in that situation where I’m constantly telling him off.*”

#### 3.1.2. Endurance

Carer 2 described frequently being confronted by the progressing apathy of her spouse through his lack of preference regarding food. She continued to ask his opinion, partly from habit but also to include him in choices, although was usually met with disinterest. As a result, the carer began to sometimes refrain from asking, in an effort to reduce opportunities to see the changes and minimise her distress. Whenever it happened, the carer described needing to be resilient and simply move on in order to cope:


*“And I still sort of ask his opinion about things because you always have done and then… just, sometimes it gets you down and sometimes you just cope with it and move on.”*


Participants also spoke of an adjustment process that enabled them to endure the changes in their spouses and continue to provide support. Carer 6 expressed that she was initially bothered about changes in behaviour, but had grown accustomed to them. The loss of initiative in her spouse required the carer to constantly prompt him to complete house chores or to leave the house to run errands, amongst other things. The carer resolved that she had “*got to take every day as it comes*” to avoid feeling overwhelmed by the need to take on new additional roles and responsibilities that had previously been her spouse’s.


*“Sometimes I just think—I’ve got that many jobs to do, I don’t know what to do. It does bother me but then again, I think, it is what it is, isn’t it? I can’t do much about it.”*


Accepting that some tasks were beyond Carer 6’s abilities, and accepting her spouse’s loss of interest in them, enabled her to move forwards and outsource such tasks, for example, hiring a gardener (*“It was taking us a while to get the grass cut and I thought, I’ve got enough to do in the house without doing this. So, I thought we’ll have a gardener”*). This ability to seek additional support enabled a sense of endurance and ability to cope, and provided time for carers to focus their efforts toward activities that benefitted them, or the person with dementia instead, but it came at the cost of acknowledging the apathy and letting it happen.

#### 3.1.3. Frustration with Lack of Insight

Most of the participants mentioned that the person with dementia lacked awareness of their apathy, and in some cases of their dementia diagnosis. This absence of insight led to frustrating scenarios; for example, Carer 6 described having to undertake redecorating on her own, which was a challenge due to her own health, only for her spouse to take full credit for her efforts.


*“Last year we did decorate the lounge and I’ve got my own health issues. But he hadn’t got a clue how to do it… I have done decorating myself. When it was done, he thought he had done it.”*


Carer 6 reported that her spouse was in denial of both his diagnosis and apathetic behaviour, and assumed that he had completed the job because he typically would have done in the past. The lack of awareness may contribute to maintaining his identity of being capable and helpful, but the discrepancy between this process and the experience of the carer led to feelings of frustration. Carer 5 also described a frustrating situation arising from a lack of insight, where her spouse had misunderstood his role and purpose for going to the day centre, and she needed to explain to him that he was attending as a client instead of a teacher.


*“…I keep saying, you’re not there to teach, they’re there to help you do whatever. So, I think it’s getting him to cross over, that term, you are-well, I call them clients and you’re not the teacher, because that’s what he likes. My husband likes to be in control. So, not having that control any more, …you’re not going to be there teaching other people, you’re not to be the control person, and how he’s going to find that, I don’t know.”*


Here, the carer felt frustrated because she thought her spouse was unwilling to accept his new role as a client, and worried that he might not behave appropriately at the day centre. The person with dementia may have maintained their perspective of attending in the role of a teacher to normalise going to the day centre, and maintain their sense of identity as someone who is in charge. In this situation, the carer had a difficult balance between letting him believe in his perspective, because this was likely to encourage him to go to the day centre, versus breaking his perspective down in order to help him behave appropriately and make things easier for the staff and other attendees of the day centre. Across these accounts, a tension between slightly deceiving the person with dementia in order to mollify their feelings versus confronting them with reality to either encourage appropriate behaviour or wrestle back the credit for completed tasks existed, and this contributed to the delicate balance that carers had to navigate between their emotions.

### 3.2. Theme 2: New Roles Imposed by Caring

Carers described that as their spouses’ dementia progressed, the extent of the apathy tended to increase, and consequently, carers found themselves taking on new and varied roles that previously were not their responsibility. The different demands required a degree of adaptation on the part of the carers, and they expressed a sense that their status as partners within their marriage was changing towards an unequal level of responsibility and power in their hands.

#### 3.2.1. Additional Responsibilities

The lack of initiative to start or complete household tasks that previously fell within the domains of the person with dementia resulted in carers doing more than their usual workload or perceived fair share of chores. Carer 3 described that her spouse would wash dishes in the past, but this had now stopped:


*“He won’t wash up now, he just doesn’t want to get out the chair and do it, so I am ending up doing the jobs that he used to do and… he did moan about them a couple of times but it wasn’t to the point that he would never do them, he would always do them.”*


Previously, Carer 3’s spouse could overcome their reticence towards this chore to complete the task, but now apathy seemed to hinder this ability to overcome the desire to not do the task. Carer 3 recognised that being unmotivated towards performing unfavourable tasks is natural, but typically a person would feel responsible or wish to avoid disappointing others and therefore would complete the task nonetheless. This was no longer observed in her spouse, and the carer attributed this to his apathy.

Carer 5 described administrative roles in the family that her spouse had always held, and consequently only he knew how to perform the associated tasks. Now, due to her spouse’s apathy towards these tasks, the carer had no choice but to quickly learn the skills in order to assume the role of her spouse.


*“I don’t do, I (have) never done car insurance. He’s always done that but now he can’t do any of that. (So) I have to go on and I don’t know. Now I kind of know what I’m doing because I have to. … I say to him, look, you just leave it, you just leave it.”*


The role of organising household administration, such as car insurance, is often an activity undertaken by men in this age group. Therefore, the change in ownership of this role represents more than just an extra task to do, but rather a move beyond traditional gendered roles that have potentially defined the divisions of labour in relationships. It demonstrates a change in the relationship from a partnership where each partner is responsible for different but equal tasks to a more managerial role for the carer. The carer also expressed that over time she had continually adopted roles and responsibilities which were previously the responsibility of her spouse, meaning that she was required to constantly master important responsibilities to keep the household afloat. Despite how overwhelming this may be, the carer showed compassion and a sense of duty towards her spouse because she assured him to leave tasks to her, recognising his change in abilities.

Although many of the carers felt that the people with dementia did not realise the extra tasks they were undertaking, Carer 6 reflected that her spouse did have an insight into her changing role and was very appreciative:


*“But when he is eating his dinner, he always says, that’s lovely and he says things like, I’m lucky to have you and you look after me and things like that.”*


#### 3.2.2. Reigniting “The Spark” by Promoting Interests

An important new role taken on by carers was that of a quasi-activities manager for their spouses, and most participants described the lengths they would take to overcome their spouses’ apathy through engaging them in various interests. Carers described this in terms of a “*spark*” or “*sparkle*” that could be elicited with the right circumstances or activities, but which became harder to generate as the apathy progressed.


*“Well, he has sparks. Yeah, he has his things that do make him spark.”*


(Carer 5):


*“It’s gradually building up to a point… where he’s not really interested in anything. … I did notice that, that the sparkle wasn’t there that he used to have when we were going away and things like that.”*


(Carer 3):

Carer 5 described the “*spark*” as a rarity, and consequently she seemed to be always on the hunt for new activities to pique her spouse’s interest (“*I’m always thinking of what I can do for him*”). At the time of the interview, the person with dementia was keen on making walking sticks, but Carer 5 explained this would be a passing phase until he lost interest in that too (“*This is just something that he has, that keeps him going, until he loses interest in that. I’ve got to think of something else*”), and she would need to find the next thing, such as “*jigsaws, and different things that we can do*”. It seemed harder and harder to identify things that would elicit a sense of enthusiasm each time, but carers felt compelled to continue the search. Carer 3 interpreted the sparkle in her spouse as a past version of her spouse, prior to the apathy, which represented the ‘real him’ somewhere deep inside, but was occluded by a layer of indifference. A “*little light*” could be elicited if only the carers could find the right activity or interest, and this juxtaposed with the darkness of apathy to create a ray of hope. Therefore, any signs of this spark motivated the carer to put in the extra effort to preserve these interests. Carer 1 also described how he had tried to reignite his spouse’s interest in reading by trying different types of books, but changes in her concentration made reading too difficult now. The carer persevered, however, and kept looking for similar activities and “*things that could get her interest*”, and was delighted upon discovering something that would return her spark.


*“Which is why I’m so pleased at the moment, … her doing word search, she does it all the time. You know, we buy them and there’s one that comes through the mail… And there’s a pile by her bedroom.”*


The efforts of the carer to constantly search for things that might generate an interest demonstrate a dedication beyond general care, but this kept the spark ablaze and ensured that they remained connected to the person underneath the apathy. Carer 6 also went above and beyond to create opportunities to elicit enthusiasm and enjoyment for her spouse; although a holiday within the United Kingdom would be convenient and less costly, she took her spouse to Disneyland in America because of his love of rides and theme parks.


*“It’s (theme parks) one thing that he really loves. So, that’s why we decided we’d take him. It’s like he hasn’t got dementia when he was there. It was really good.”*


Despite the knowledge that he would lose his memories of the holiday, the carer was unquestioningly prepared to spend time and money, and to navigate logistical challenges to see the spark in her spouse again. The lengths taken to achieve this were worth it for the carer, because knowing her spouse had had such a good time, and was present in that moment, could carry them through even after the memories had faded.

### 3.3. Theme 3: Having a Life of One’s Own

Carers recognised that although their focus and role had shifted over time from that of an equal partner to the role of carer, taking care of themselves was an important strategy to cope with the challenges that apathy created.

#### 3.3.1. Pursuing Own Hobbies and Interests

Pursuing hobbies and interests was difficult for some carers, but they recognised the need and value of making time for their own interests and enjoyment. Carer 4 reflected that she would remove herself from situations that her spouse’s apathy had created, such as watching the same television programmes repeatedly, and this gave her the space to engage with her own hobbies.


*“So I tend now to just put myself in another room because he’s just sitting there watching the same programmes over and over again, which does get me down”.*



*“Sometimes I feel quite motivated because I like sewing and I like knitting and I feel quite motivated.”*


However, Carer 4 acknowledged that she needed “*to be in a relaxed mood before I can start*” to benefit from her hobbies, and may have wanted to avoid associating feelings of stress with activities that provided relaxation and escape to preserve them as a relief from the caring role. Her spouse also sometimes made it difficult for her to engage in her hobbies by wanting to know where she was, and she wondered if “*he’s frightened to be on his own perhaps*”. Carer 1 described a similar situation where his spouse began “*following me around the house almost, yes. To sort of make sure I’m there, where I am.*” This made it hard for Carer 1 to find the time and space to undertake activities for themself.

#### 3.3.2. Finding People to Talk To

Carers received different forms of support from others, which provided them with both an outlet for their feelings, and time away from the person with dementia. Carer 4 reflected that connecting with friends provided her with social interactions that no longer existed at home.


*“Luckily I’ve got some friends here so I’ve got somebody to talk to. And get what’s happening in their lives as well, not just (being in) my little bubble.”*


For Carer 4, communication with her spouse was challenging throughout their marriage, and had become more difficult due to the apathy. Talking to others outside the household and hearing about their lives enabled Carer 4 to break out of her “*bubble*” and engage with the wider world. Interacting with other carers in similar situations provided a forum to share experiences and learn from each other.


*“And it also helps when you go to places like Song Time; you’re talking to the other carers. Also, I go to the Alzheimer’s meeting once a month as well and again, you’re talking to the other people and it just helps.”*


The various activities and groups benefitted the person with dementia, but also the carer as well; finding out about what the future might hold and sharing helpful strategies made her feel less alone and more confident.

Carer 3 described how her children could act as a form of support:


*“My daughter will come round and say, go on bugger off and do what you want to do, I will sort him out don’t worry and she does bless her.”*


Family members provided a break from caring to enable carers to spend time on their chosen activities. Support from close relatives gave the carer confidence because they are trusted and known individuals, rather than hiring someone unfamiliar to care instead. Further support came from formal sources such as clinical services sought by the carers to help with the emotional and stressful aspects of the role. Carer 6 revealed:


*“I had to go for some counselling and I’m on anti-depressants.”*


Caring for someone with dementia can result in low mood and anxiety, and this is made more challenging with the added impact of apathy, and in some circumstances formalised support was necessary to be able to cope.

## 4. Discussion

This study aimed to explore the impact of apathy in dementia on carers and their understandings of how apathy manifested in the person for whom they care. The findings extend existing knowledge of this area by providing an insight into carers’ lived experience, and demonstrate that apathy in dementia does impact carers. To the best of our knowledge, this is the first study exploring the lived experiences of carers in response to apathy in dementia. Carers described apathy as a change in behaviour, personality, and interest, specifically a lack of initiative in starting or completing daily activities. This included not participating in usual responsibilities around the house, such as household chores, and unwillingness to leave the house. The accounts of apathy were in line with previous descriptions of apathy as a loss of motivation in relation to previous level of functioning [1].

Carers went to significant lengths to sustain hobbies and interests in their spouses and described this as keeping their “spark” ignited. The way the carers spoke about the spark implied that they knew it was still alight, somewhere deep within the person with dementia, and as long as they could keep finding things to fuel, the spark it would keep burning. The spark seemed to represent the essence of the person with dementia from before their apathy, or even before they had dementia, and represented an essential link between past and present that helped carers and their spouses maintain a connection.

Previous studies suggest that neurological changes cause the lack of motivation [4,5,6], but this study suggests that other factors may also influence the development or protection from apathy [7,8]. The efforts and interactions of the carers to reignite this “spark” in their spouses enabled people with dementia to overcome the lack of motivation through constant stimulation and encouragement to participate in various activities. Viewing apathy as simply the result of neurological changes limits the scope for interventions, whereas considering apathy as the result of environmental and social factors provides people with dementia and those around them with agency to intervene and make a difference in their lives, particularly with the use of non-pharmacological therapies [29].

The carers’ efforts support our understanding that family carers are a vital determinant of positive outcomes for people with dementia [30]. For instance, previous research has shown a 20-fold protective effect on the risk of institutionalisation when the person with dementia had a coresident carer [31], and remaining at home is associated with better quality of life for both people with dementia and carers [32]. Dealing with apathy is challenging and distressing for carers, but this research suggests that it is valuable to prevent further negative impacts.

### 4.1. Impact on Carers

Previous studies investigating the impact of apathy on carers of people with PPA and ALS [12,13] used the Zarit Burden Interview [33], a quantitative approach, to measure carer burden. This provides a narrow perspective of burden and does not allow for exploration of the subjective experience of the carers [34]. In the present study, each carer could share their lived experiences and their interpretation of what was happening to both themselves and the person for whom they cared [35]. Simpson et al. [16], exploring apathy in PD, found similar insights regarding the social context of apathy, showing the interaction between the individual’s apathy and other people in their lives.

A recent systematic review [30] identified that demands of caring were related to the quality of life for family carers. All studies included in the review reported a negative association between quality of life and subjective carer burden. In our study, carers reported that the impact of apathy created new roles and additional responsibilities as they took over tasks previously held by the person with dementia. Carers’ responsibilities were further increased through promoting interests and hobbies of their spouses. Less initiation of activities or household tasks by the person with dementia required the carer to compensate for the loss of a more equal partnership and adjust to a caring role. Our findings support the systematic review, and suggest that apathy operates as a mechanism of action in this association, whereby the apathy in a person with dementia creates additional responsibilities for carers, contributing to increased burden on the them and, in turn, reducing their quality of life.

The different strategies and tactics employed by carers to overcome their spouses’ apathy, such as taking them out without seeking their opinion, or withholding details about it, created a complex set of emotions. Although the ends apparently justified the means, because this was for the benefit of the person with dementia, carers felt guilty for misleading or withholding information from their spouse. Studies in the literature concerning guilt for carers of people with dementia show that guilt is strongly associated with carer burden, and is related to a loss of closeness in their relationship with the person with dementia [14]. Our findings suggest that carers tended to find themselves in a catch-22 situation, whereby they were either committing deception and trickery to engage the person with dementia in an activity or outing, or doing nothing, with the result that the person with dementia would be unoccupied and unengaged—both evoked feelings of guilt and sadness. These conflicting emotions were experienced even when such strategies delivered positive outcomes for both the carers and their spouses. Understanding the perspective of the person with dementia could help to alleviate feelings of guilt by providing reassurance that their spouses genuinely appreciated their efforts because they were desperate to do something, and required this kind of help to initiate it or overcome barriers [20]. Counselling could help couples to foster empathy because it allows forgiveness and lowers feelings of guilt and shame [36].

Carers discussed their frustration at the frequent lack of insight in the person with dementia about their apathy. This tallied with Pfeifer et al. [17], highlighting a discrepancy between the carers’ and the person with dementia’s perspectives of the manifestation of apathy. In our other publication focusing on the experiences of people with dementia from their perspective [20], it appeared that lack of insight may represent an internal struggle to maintain self-identity, and serves an important purpose in providing continuity in the sense of self through a period of changing abilities. From the perspective of the carers, this invisible struggle is unknown, and therefore difficult to understand the reasons that underpin it to render it less frustrating. If carers have an understanding of this perspective, it could possibly ease their annoyance when faced with apathy, and thereby reduce their sense of burden or relationship strain.

### 4.2. Strengths and Limitations

This is the first study to explore the lived experience of apathy in people with dementia from the perspective of their carers. The sample size was optimal for an IPA study [24], and interview data were rich, because most participants shared in-depth accounts of their experiences. Analysis was conducted by M.C. and W.B., and developed and refined through discussions involving all four researchers. All interviews began by asking participants to describe their understanding of the term apathy, and how they recognised it in the person for whom they cared. This enabled the research team to be confident that participants were interpreting situations and events through the lens of apathy, even if they used their own terminology, and it helped participants to feel comfortable discussing apathy in a way that was meaningful to them.

Some limitations must be acknowledged. Participant recruitment was limited to the local area to enable visits to local groups, and for ease of participants travelling to the university to participate. Organisers of local groups were gatekeepers to group members, and in some cases prevented access over concerns that members were becoming over-researched, and that the purpose of support groups is not to provide research participants.

The participants with dementia had a limited range of diagnoses, which may not have encapsulated different types of dementia, because diverse types may impact carers in different ways. This was an IPA study; therefore, the focus of the approach was to capture the experience of each participant on an idiographic level [22] and we do not seek to generalise our findings to all carers, caring for people with all types of dementia. This study articulates the lived experience of this sample, and highlights potential implications that may provide useful insights to be investigated further.

### 4.3. Implications

#### 4.3.1. Recommendations for Carers

Support networks for carers are vital to help them cope with the emotional impact of apathy, find time for themselves, and share ideas about how to engage the person for whom they care. This support could come from family and friends, but external support such as carers groups are also particularly useful in enabling carers to share experiences and learning with each other, and consequently, awareness of local groups and events is important. Many carer groups have an online presence, and empowering carers to navigate the internet with confidence may enable a greater awareness of, and engagement with, such groups. There is also the potential for the development of community and residential care, specifically focused on apathy. Currently, general carer support exists in well-established community services, such as support groups [37], singing, music and art groups [38], but groups specific to apathy are lacking.

#### 4.3.2. Recommendations for Clinical Practice

There are a lack of services and therapeutic interventions within health and social care that focus on apathy management for either people with dementia or carers. Our study suggests that carers face initial difficulties in accepting changes in the person for whom they care related to apathy, but reflection and focusing on the bigger picture when deploying particular engagement strategies is helpful. Encouraging acceptance as a coping style could be advantageous in providing a positive adjustment to such challenges [39]. Therefore, acceptance-based therapies, such as Acceptance and Commitment Therapy or Dialectical Behavioural Therapy [40], could be valuable in maintaining wellbeing for carers [41]. The carers in this study had no difficulty understanding the term apathy, and were able to describe in their own words. Using this approach in clinical encounters could open up the discussion to include a more complete discussion of apathy which is relevant to carers.

#### 4.3.3. Recommendations for Policy

Dementia cafes were launched in the United Kingdom in the year 2000 to promote opportunities to connect with others and share experiences, receive formal support from experts, and learn from each other [42,43]. These cafes are typically referred to as “memory cafes”, because the general focus is on memory changes and impairments. This is reflective of the wider conceptualisation of dementia as a memory problem; memory changes tend to be the symptom noticed or discussed first, with less awareness of non-memory symptoms [44]. Despite the substantial impact of apathy, it is not addressed or discussed in the same way as other symptoms, perhaps due to a lack of interventions or forums that address this particular symptom. Given the evidence to demonstrate the negative impact of apathy on carers, the creation of “apathy cafes” to spread awareness and support carers and people with dementia who experience apathy may be useful.

#### 4.3.4. Recommendations for Research

Future research must explore carers’ perspectives of apathy in dementia in Black, Asian, and minority ethnic communities. Different communities and cultures may experience and manage apathy in different ways; for example, in cultures where an obligation for adult children to care for older family members, in line with the Asian value of filial piety [45], is the norm. Filial piety can be experienced as a societal pressure or a willingness to care, and differing perceptions of this could lead to higher levels of carer burden [46]. Hence, caring for a person with dementia with apathy may expose some communities to increased carer burden. Research must also consider how family carers from a variety of cultural backgrounds interact with health, social, and third sector services to support care for people with dementia who have apathy. The prevailing stereotype that family carers from Black, Asian, and minority ethnic communities are capable of caring for their older relatives with support from large intergenerational and extended families [47] may not always be the case. Apathy may remain unnoticed due to a natural transfer of household responsibilities as a mark of respect for older family members, who are no longer asked or obliged to participate in chores or activities as these are taken up by younger generations. A more inclusive understanding of apathy in dementia and the support needs of carers across different communities is required for equitable progress.

## 5. Conclusions

Through exploring the lived experience of carers for people with dementia who have apathy, this study found that apathy created conflicting emotions, led to carers taking on new roles and responsibilities, and that respite from the caring role can be achieved through pursuing activities of their own. Apathy in people with dementia can place emotional and physical demands upon carers, who find themselves employing a variety of strategies in order to engage the person for whom they care. Understanding the experience of carers provides opportunities to consider how apathy is conceptualised and discussed at a societal level, and how existing interventions may be utilised to alleviate carer burden. Supporting carers to understand the experience of person with dementia with apathy could help to minimise their sense of guilt and frustration, and increase a sense of acceptance.

## Data Availability

Data are held by the research team.

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
