# Peer review of "“He Just Doesn’t Want to Get Out of the Chair and Do It”: The Impact of Apathy in People with Dementia on Their Carers"

_ijerph, 2021, doi:10.3390/ijerph18126317_

Round 1
Reviewer 1 Report
Thank you for this brilliant paper. The paper has explored an important topic about the carers’ experiences in caring for people with dementia who have apathy. Apathy is common clinical issues and can cause the carers a great deal of burden. Qualitative methods seem appropriate to obtain deeper understanding around the topic. The findings are very interesting and could inform clinical practice and policy. However, I have some suggestions to improve the paper, especially for the discussion part and would encourage the authors to consider and amend the paper accordingly.
- P1 line 39: Caregivers or carers - please be consistency. Please recheck across the paper.
- P1 line 44: What are the differences to the carer burden in the previous literature mentioned in above paragraph? Please make the gap clearer. What are aspects of the impact of apathy in dementia on carers that is lacking for research? The authors may also wish to combine this to the 5th paragraph mentioning the needs of qualitative research.
- P2 line 51-52: ‘…it is the carer’s reaction that directly influences the BPSD-related distress’. Whose distress? Do you mean carers' (their own) or the person with dementia’s distress?
- P2 line 67: “Due to the stability of apathy over time…”. It’s unclear to me that if the apathy is stable, not progressing, it will have negative consequences or be associated with poor outcomes?
- P3 line 108 and Sampling and recruitment overall: “…,or whether the carer 107 or person with dementia explicitly self-identified as experiencing apathy…”. It can be clearer what clauses or wordings that the authors have used when recruiting the potential participants. How did the authors identify the potential participants? Did the authors approach to PLWD or their carers? How the potential participants know if they would have, or they care for people with apathy? And how can the authors be sure upfront that they would have the care experience or burden resulting from apathy symptoms? I guess the carers would generally not be familiar with the term and symptoms of apathy. I also guess the person with dementia with apathy would be less able to recognise the symptoms or disinterest in taking part. Some insight around the recruitment strategy would justify this study’s recruitment and benefit future studies.
- From results – P4 line 188-189: Some discussion about impact of role reversing would be interesting.
- P4 line 192-195: This is important insight and findings on how difficult the balancing conflicting emotions is.
- P6 line 249-250: It's a bit strange to me that the person with dementia would intentionally want to get the credit. Maybe rephrase to something that the person thought she/he is still able do something by themselves (unable to acknowledge or unconcern about his/her limited abilities).
- P7 line 299, P8 line 376 (also please recheck and use the participants’ wordings across the findings or add some explanation to the use of this term at the beginning of results): Is it the exact term the participants used? It's quite interesting to see how apathy was introduced by the interviewers to the participants during the interview and why the participants attributed this to apathy. I guess this term would have not been widely known by the carers, or at least not every participant in this study would be familiar with the term from the outset. Also, it would be really interesting to know how they perceived and actually called the symptoms in the first place. I think the authors have mentioned something around this in the introduction paragraph of discussion (P 9 line 427-428), but this should come as part of results. Then in the discussion the authors may wish to discuss further on how the introduction of concept around apathy by the researchers to the carers would influence the interviews. This may inform clinical practice on how we can discuss such symptoms with the carers. And it would indicate how carers in general are aware of such problems.
- P7 line 307-317: I think most of this paragraph should go with the discussion as it's more interpretation and should not have been the participants' views or experiences (i.e., not the findings).
- From theme 3.3.2 Finding people to talk to: Apathy could affect their relationship and may further cause loneliness or social isolation in the carers as the person and their carer may live by themselves, separately from other family members. Some discussion around this issue and implication/intervention would be really interesting.
- Discussion: It can be slightly off-topic. But it is really challenging to the carers or healthcare professionals to provide patient-centred care if the person has apathy and cannot express or don't have interests in things. Some discussion around the patient-centred care for this specific population would be interesting.
- P10 line 440-448: It's interesting discussion but can be expanded or be more specific on how we can do this or what existing non-pharmacological interventions can be applied for this.
- P10 line 457-464: It should be more specific to what kinds of those terms like subjective experiences, burden, social context and insights that are in line with the previous studies - depression, social isolation, frustration or what.
- P11 line 542: This part is quite unclear. Please explain a bit more about what the bigger picture is (overall dementia trajectory? or overall care that the carers have provided?).
- P11 line 543-547: Please state clearly that theses interventions are for the carers.
- Recommendations for research: Social cultural and ethnicity can hugely impact the perception of carers and the care for the person with apathy. And it's really important to draw social, cultural and ethnicity background to the future research, but this can be more succinct (one paragraph would be sufficient). Also, it seems a bit jumpy because it has not been directly come up from the current study. Suggestions for future research can be expended to cover other aspects and built on the limitations or highlighted findings of the current study.
- Typos: P5 line 230 ‘the him’, also I’m not if it is due to typesetting, but the authors should recheck subthemes for 3.2.3, 3.2.4, 3.3.3.
Author Response
We are thoroughly grateful to you for the consideration and detail that you have provided, and we were very pleased to read that you felt the paper addressed an important topic. Please see the table below for a point by point reponse to your comments.
|
Thank you for this brilliant paper. The paper has explored an important topic about the carers’ experiences in caring for people with dementia who have apathy. Apathy is common clinical issues and can cause the carers a great deal of burden. Qualitative methods seem appropriate to obtain deeper understanding around the topic. The findings are very interesting and could inform clinical practice and policy. However, I have some suggestions to improve the paper, especially for the discussion part and would encourage the authors to consider and amend the paper accordingly.
|
Thank you for your acknowledgement of our work in this area, we are delighted to hear that you found it interesting and informative. We are very grateful for your detailed and helpful suggestions which we have thoroughly considered and feel they have improved the paper |
|
P1 line 39: Caregivers or carers - please be consistency. Please recheck across the paper.
|
Thank you for highlighting this. We have now amended for consistency and used the term carers. |
|
P1 line 44: What are the differences to the carer burden in the previous literature mentioned in above paragraph? Please make the gap clearer. What are aspects of the impact of apathy in dementia on carers that is lacking for research? The authors may also wish to combine this to the 5th paragraph mentioning the needs of qualitative research.
|
Thank you for pointing this out. We have amended the text as shown.
Originally: BPSD generally have a negative impact on carers including stress, inability to cope with accumulating tension, desperation, exhaustion over time, isolation, and relationship issues between the carer and person with dementia (10, 11). Research that directly explores the impact of apathy in dementia on carers is lacking, although studies have explored this in other conditions.
Revised to: BPSD generally have a negative impact on carers including stress, inability to cope with accumulating tension, desperation, exhaustion over time, isolation, and relationship issues between the carer and person with dementia (10, 11). However, the effects of different BPSD are often lumped together and research that directly explores the impact of apathy in dementia on carers is lacking. Some published studies have explored this in other neurological conditions. For example, apathy in PPA etc etc
|
|
P2 line 51-52: ‘…it is the carer’s reaction that directly influences the BPSD-related distress’. Whose distress? Do you mean carers' (their own) or the person with dementia’s distress?
|
It is the carer’s distress and we have added the word carer’s to indicate who is distressed |
|
P2 line 67: “Due to the stability of apathy over time…”. It’s unclear to me that if the apathy is stable, not progressing, it will have negative consequences or be associated with poor outcomes?
|
The point we were making is that apathy has negative consequences for carers because it doesn’t improve – due to the discrepancies in evaluations of apathy it is difficult to know if it does get worse per se, but ultimately the enduring nature of it is challenging. However, this paragraph was unclear, and so the first part was removed and the second part was combined with the following paragraph to make a stronger point about why research with carers is important. |
|
P3 line 108 and Sampling and recruitment overall: “…,or whether the carer or person with dementia explicitly self-identified as experiencing apathy…”. It can be clearer what clauses or wordings that the authors have used when recruiting the potential participants. How did the authors identify the potential participants? Did the authors approach to PLWD or their carers? How the potential participants know if they would have, or they care for people with apathy? And how can the authors be sure upfront that they would have the care experience or burden resulting from apathy symptoms? I guess the carers would generally not be familiar with the term and symptoms of apathy. I also guess the person with dementia with apathy would be less able to recognise the symptoms or disinterest in taking part. Some insight around the recruitment strategy would justify this study’s recruitment and benefit future studies.
|
Thank you for this important point. A member of the research team visited local community support groups to speak about the study and then had conversations with group members who wished to know more and were interested to take part. As we have stated on line 113 we did not require participants to explicitly state that they had apathy or that they cared for someone with apathy and as such we were not sure upfront that the participants would have the care experience or burden resulting from apathy. To ensure that we were sure participants recognised and understood the term, we began each interview by asking them to explain what it meant to them and if they recognised it in their daily lives. Based on our previous experiences of research and clinical practice with people with dementia and carers we had a good insight into the lived experience of caring for someone with dementia and felt confident that approaching attendees of local community groups would enable us to recruit participants who could give us a detailed insight. We have amended the text on lines 106-109 accordingly. |
|
From results – P4 line 188-189: Some discussion about impact of role reversing would be interesting.
|
Whilst we do agree, and potentially could discuss the results for many, many pages, we must present a concise account to avoid the paper becoming overly verbose. |
|
P4 line 192-195: This is important insight and findings on how difficult the balancing conflicting emotions is.
|
As before, we are very much in agreement here regarding this important insight. |
|
P6 line 249-250: It's a bit strange to me that the person with dementia would intentionally want to get the credit. Maybe rephrase to something that the person thought she/he is still able do something by themselves (unable to acknowledge or unconcern about his/her limited abilities).
|
It certainly is a strange account and from conducting the original interview, through the analysis, and on to writing the publication it was an event that struck us, but we must keep in line with the principles of IPA and keep close to the data and the participant’s interpretation of it. |
|
P7 line 299, P8 line 376 (also please recheck and use the participants’ wordings across the findings or add some explanation to the use of this term at the beginning of results): Is it the exact term the participants used? It's quite interesting to see how apathy was introduced by the interviewers to the participants during the interview and why the participants attributed this to apathy. I guess this term would have not been widely known by the carers, or at least not every participant in this study would be familiar with the term from the outset. Also, it would be really interesting to know how they perceived and actually called the symptoms in the first place. I think the authors have mentioned something around this in the introduction paragraph of discussion (P 9 line 427-428), but this should come as part of results. Then in the discussion the authors may wish to discuss further on how the introduction of concept around apathy by the researchers to the carers would influence the interviews. This may inform clinical practice on how we can discuss such symptoms with the carers. And it would indicate how carers in general are aware of such problems. |
The participants themselves don’t explicitly use the word apathy, but we did begin each interview by finding out what their understanding of apathy was, and they were all familiar with the term and could describe it. Where we have used this term, it is based on an interpretation of larger sections of the transcript than just the quote supplied (it wouldn’t be practical to include large sections in the paper of course), and so we are confident that the participants are talking about apathy – and further to that a part of IPA is the role of the double hermeneutic, where the researcher makes sense of the participant’s own interpretation, and that is the process used here to arrive at this terminology. The beginning of the results section does include insights into how the participants described apathy, which we feel clarifies this, and approaching the interview by asking participants to explain their understanding of apathy was a helpful way to start. We are very grateful to you for raising the idea that this may inform clinical practice, and as such we have added to the strengths section, and the recommendations for clinical practice section to elaborate on this further, as we would be very pleased if our work was able to empower clinicians to discuss apathy more. |
|
P7 line 307-317: I think most of this paragraph should go with the discussion as it's more interpretation and should not have been the participants' views or experiences (i.e., not the findings).
|
Thank you for your comment but we must disagree as this is an IPA study and it is within the guiding principles of IPA to provide interpretation and contextualisation along with the description of the findings in the results section. |
|
From theme 3.3.2 Finding people to talk to: Apathy could affect their relationship and may further cause loneliness or social isolation in the carers as the person and their carer may live by themselves, separately from other family members. Some discussion around this issue and implication/intervention would be really interesting.
|
The data regarding this finding were certainly present, but not voluminous enough to prompt a more thorough discussion of this issue. We do agree that it is an important point, and one that will likely be explored in much greater depth in the future due to the pandemic restricting socialising for many months. For brevity, we would prefer not to add further discussion points in to this paper as it is already quite substantial. |
|
Discussion: It can be slightly off-topic. But it is really challenging to the carers or healthcare professionals to provide patient-centred care if the person has apathy and cannot express or don't have interests in things. Some discussion around the patient-centred care for this specific population would be interesting.
|
We completely agree and acknowledge that simply talking about apathy is difficult in the first place, never mind helping someone to express themselves further. Some insights that might be helpful on this particular topic are addressed in the companion paper to this one, which explores the experience of apathy from the perspective of people with dementia, which you might enjoy reading. |
|
P10 line 440-448: It's interesting discussion but can be expanded or be more specific on how we can do this or what existing non-pharmacological interventions can be applied for this |
We do cover potential non-pharmacological interventions in section 4.3.2 and in order to keep the paper from becoming extremely long we would prefer not to expand on this at this time. It is certainly something we wish to explore in the future. |
|
P10 line 457-464: It should be more specific to what kinds of those terms like subjective experiences, burden, social context and insights that are in line with the previous studies - depression, social isolation, frustration or what.
|
It may seem disrespectful for us to say so but this comment is difficult to understand and we don’t see the point that is being made.
We have reviewed the paragraph and we feel that it does make the points that we wish to convey: that is, the ZBI as a quantitative measure does not tell us much about people’s experiences, whereas our study and that of Simpson et al both provided insights into the interactions between carers and pwd around apathy. |
|
P11 line 542: This part is quite unclear. Please explain a bit more about what the bigger picture is (overall dementia trajectory? or overall care that the carers have provided?).
|
The bigger picture here refers to the greater good for the person with dementia. By acknowledging that the person with dementia does benefit from the strategies used by the carer, even if they involve deception, carers can absolve themselves of some of the difficult feelings by knowing that overall the person with dementia is having a positive experience because of their efforts. |
|
P11 line 543-547: Please state clearly that theses interventions are for the carers.
|
Thank you for suggesting this clarification, we have amended the text accordingly. |
|
Recommendations for research: Social cultural and ethnicity can hugely impact the perception of carers and the care for the person with apathy. And it's really important to draw social, cultural and ethnicity background to the future research, but this can be more succinct (one paragraph would be sufficient). Also, it seems a bit jumpy because it has not been directly come up from the current study. Suggestions for future research can be expended to cover other aspects and built on the limitations or highlighted findings of the current study. |
We agree, this paragraph does cover perhaps more material than is needed to make the point, and as such we have edited it to make it more succinct. |
|
Typos: P5 line 230 ‘the him’, also I’m not if it is due to typesetting, but the authors should recheck subthemes for 3.2.3, 3.2.4, 3.3.3. |
Thank you for noticing this – the ‘the’ is an error and we have removed it. Thank you for alerting us to these subthemes as well, it is indeed a typesetting issue. |
Reviewer 2 Report
Thank you for recommending me as a reviewer. This paper aimed to explore the lived experience of apathy in dementia from the perspectives of the people with dementia and their carers. This qualitative study is of academic significance. If the authors complete minor revisions, the quality of the study will be further improved.
- If the authors describe more specifically the definition of apathy and the neurological mechanism (theoretical background) in the Introduction section, this may help readers understand.
2. line: 88-96. In qualitative research, data collection methods, environment, and research procedures are important. If the authors describe the Qualitative Research Methods (IPA) more specifically in the Methods section, it can help readers understand.
3.line 560-: This study includes "Recommendations for research". This is very desirable for subsequent researchers.
Author Response
Thank you for your insighful comments which have given us plenty to think about when further improving our paper. We have included our point by point response to each comment in the table below:
|
If the authors describe more specifically the definition of apathy and the neurological mechanism (theoretical background) in the Introduction section, this may help readers understand.
|
Thank you for your comment. To avoid the paper becoming overly long, we would prefer not to add further content to this section, particularly as a neurological explanation of apathy would not fit particularly well with our findings regarding the experience of apathy. Interested readers are most welcome to follow up on the research we have cited. |
|
line: 88-96. In qualitative research, data collection methods, environment, and research procedures are important. If the authors describe the Qualitative Research Methods (IPA) more specifically in the Methods section, it can help readers understand. |
This is a good point, and whilst we do agree that championing the role of qualitative research is important and achieved by thorough explanation of methods, we do understand that the majority of readers will be familiar with the technique or can appraise themselves of it via the references we have cited. As such, and to remain succinct, we have nothing additional that needs to be added here. |
|
line 560-: This study includes "Recommendations for research". This is very desirable for subsequent researchers. |
Thank you so much for such a positive comment. We very much agree, and hope other researchers will feel suitably inspired. |
Reviewer 3 Report
Relevant paper, to increase our knowledge of impact of apathy of persons with dementia on their informal caregivers. Alas, the sample is very small (n=6) and not clearly described with a table of respondent characteristics. Also, the type/stage/duration/etc of dementia in the spouse of "demented" partner stays unclear so that the care situation remains a black box. So I doubt the authors choice to split this research in two separate papers for de persons with dementia (reference 20) and this paper on garegivers, both with very small samples. This major issue made me almost reject the paper, so you must elaborate much stronger on the relevance of this research!
Another rather unclear issue to me is what the scope of "impact" of "apathy" is in this paper. Is seems to be mixed with other themes like "insight in the disease", "communication issues", "passive anger" at both sites of patient and caregiver. Related to but not the same as "apathy", further mixed up with vague line between "impact" and coping, and underlying themes like guilt and frustration at the site of the caregiver.
Minor points are:
- it's unclear to me when you cite quotations in italics and when not.
- please do not talk about "black communities" but use more appropriate terms like Anglo-African communities.
- your interesting findings about "the spark" me deserve more elaboration and reflection in the Discussion, and also in the Abstract.
Author Response
Thank you for your comments, which have given us much to consider. We are very pleased to read that you felt the findings were interesting, and we hope the readers will too. Please find our point by point response to your comments in the table below:
|
Relevant paper, to increase our knowledge of impact of apathy of persons with dementia on their informal caregivers. Alas, the sample is very small (n=6) and not clearly described with a table of respondent characteristics. Also, the type/stage/duration/etc of dementia in the spouse of "demented" partner stays unclear so that the care situation remains a black box. So I doubt the authors choice to split this research in two separate papers for de persons with dementia (reference 20) and this paper on garegivers, both with very small samples. This major issue made me almost reject the paper, so you must elaborate much stronger on the relevance of this research! · |
We are very glad that you did not reject the paper, as it would have been a shame to reject it based on an inaccurate understanding of sample size. Hopefully, in the methods section, you will have been able to find the explanation that this is an IPA study, and you are most welcome to follow up on the references we have cited. For your interest, the desired participant number for an IPA study is between 3and 6 participants, so as you can see, we are very much at the upper end of this sample size. In fact, between the two studies having 12 participants, one might consider this sample size to be quite large. We can thoroughly recommend the work of Jonathan Smith and colleagues in understanding the IPA methodology if you are unfamiliar with this technique. We would also urge you not to refer to people as ‘demented’ as this is a stigmatising term that is not found to be appropriate by people with dementia. You may wish to use terminology such as people with dementia, people living with dementia, or people with lived experience of dementia. Further, we have clearly explained the relevance of the research, and feel that in line with other reviewers’ positive comments there is nothing further to add here. |
|
Another rather unclear issue to me is what the scope of "impact" of "apathy" is in this paper. Is seems to be mixed with other themes like "insight in the disease", "communication issues", "passive anger" at both sites of patient and caregiver. Related to but not the same as "apathy", further mixed up with vague line between "impact" and coping, and underlying themes like guilt and frustration at the site of the caregiver. · |
The Oxford Dictionary defines impact as ‘a marked effect or influence’ and we feel that the results of this analysis and indeed the issues that you have raised do indeed come under this definition. |
|
· it's unclear to me when you cite quotations in italics and when not.
|
This is a typesetting issue from the journal, and we have asked that all quotations are provided in italics if possible. We must adhere to house styles and so it is potentially not in our gift to change it, but we are grateful to you for raising this. |
|
· please do not talk about "black communities" but use more appropriate terms like Anglo-African communities.
|
We have not talked about black communities. The term used is ‘black, Asian, and minority ethnic communities’ which is the preferred term to promote inclusion. |
|
· your interesting findings about "the spark" me deserve more elaboration and reflection in the Discussion, and also in the Abstract.
|
We are delighted that you think the findings about the spark interesting, and we also agree that these are fascinating. We thoroughly enjoyed analysing this aspect of the data. However, in the interests of brevity, and to avoid the paper becoming unwieldy, we do feel it is best to leave it as it is this time. We will perhaps explore this in future research now that you have confirmed to us that this is of particular interest. |
Round 2
Reviewer 1 Report
Thank you very much for the revision. The authors have provided clear and firm responses and amended the manuscript accordingly. Therefore, I have no further comments or questions. Thank you. Good luck.
Reviewer 3 Report
I still have my doubts of the word black in lines 566 and 576, but I'll leave that to the editors of the journal.